# Preliminary Study on the Formation Mechanism of Malformed Sweet Cherry (*Prunus avium* L.) Fruits in Southern China Using Transcriptome and Metabolome Data

**DOI:** 10.3390/ijms25010153

**Published:** 2023-12-21

**Authors:** Wangshu Zhang, Yue Xu, Luyang Jing, Baoxin Jiang, Qinghao Wang, Yuxi Wang

**Affiliations:** 1Ningbo Innovation Center, Zhejiang University, Ningbo 315000, China; 17858804095@163.com (Y.X.); 2020881045@zwu.edu.cn (B.J.); wqh15657811121@163.com (Q.W.); f265a519@163.com (Y.W.); 2National & Local Joint Engineering Laboratory of Intelligent Food Technology and Equipment, College of Biosystems Engineering and Food Science, Zhejiang University, Hangzhou 310058, China

**Keywords:** sweet cherry, gibberellin, deformed fruit, UPLC-MS/MS, transcriptome data

## Abstract

Gibberellin (GA) is an important plant hormone that is involved in various physiological processes during plant development. Sweet cherries planted in southern China have always encountered difficulty in bearing fruit. In recent years, gibberellin has successfully solved this problem, but there has also been an increase in malformed fruits. This study mainly explores the mechanism of malformed fruit formation in sweet cherries. By analyzing the synthesis pathway of gibberellin using metabolomics and transcriptomics, the relationship between gibberellin and the formation mechanism of deformed fruit was preliminarily determined. The results showed that the content of GA_3_ in malformed fruits was significantly higher than in normal fruits. The differentially expressed genes in the Kyoto Encyclopedia of Genes and Genomes (KEGG) pathway were mainly enriched in pathways such as “plant hormone signal transduction”, “diterpenoid biosynthesis”, and “carotenoid biosynthesis”. Using Quantitative Real-Time Reverse Transcription PCR (qRT-PCR) analysis, the gibberellin hydrolase gene *GA2ox* and gibberellin synthase genes *GA20ox* and *GA3ox* were found to be significantly up-regulated. Therefore, we speculate that the formation of malformed fruits in sweet cherries may be related to the accumulation of GA_3_. This lays the foundation for further research on the mechanism of malformed sweet cherry fruits.

## 1. Introduce

The sweet cherry (*Prunus avium* L.), belonging to the Rosaceae family, is native to Europe and Western Asia [1]. Sweet cherries are rich in nutrients, such as anthocyanins and flavonoids, and because of their bright color, rich taste, and other characteristics, they are widely enjoyed [2]. Sweet cherries in China are generally grown in the north because the unique climate and environmental factors in the south make it difficult for sweet cherries to bear fruit. In response to this phenomenon, farmers have started spraying GAs, and while the fruit-setting rate of the sweet cherry has been significantly improved [3], there has been an increase in the prevalence of malformed fruit. The formation of malformed fruits has adversely affected the economic benefits of fruit sales, but the molecular mechanism behind this phenomenon is still unclear.

GA is an important plant hormone and signaling pathway factor that plays an important role in plant growth and development and stress resistance environment [4]. Studies have shown that GA treatment can induce parthenogenesis to increase the fruit-setting rate and thus increase fruit yield. The treatment of citrus with GA_3_ during flowering can stimulate the division of ovary wall cells and increase the size of the ovary, thereby improving the fruit-setting rate, while paclobutrazol (GA_3_ biosynthesis inhibitor) inhibits ovary wall cell division and reduces fruit setting [5]. Spraying grape ear with 30 mg/L GA_3_ 10 days before flowering can increase the sugar content in the ovary and the activity of acid invertase, especially the activity of vacuole invertase, so as to improve the fruit-setting rate [6]. Spraying 500 mg/L GA_3_ at the flowering stage of loquat can help promote fruit setting by increasing auxin synthesis [7]. In sweet cherry, GA_3_ improves the fruit-setting rate by inducing parthenocarpy and promotes the expression of cytoskeleton and cell-wall-related genes to increase fruit size [8]. Similarly, applying GA_3_ to apples before flowering can induce parthenogenesis and increase fruit setting [9]. In addition, the application of GAs on cucumbers, tomatoes, lemons, etc., can also induce parthenocarpy [8].

GAs can also affect fruit quality. Spraying with GA_3_ at flowering time increased the fruit weight and thickened the waxy layer of the fruit, effectively prolonging the storage period of the apple fruit [10]. The fruit quality, juice yield, titratable acidity (TA), and soluble solid content (TSS) of lemons after GA treatment were improved [11]. When 50 and 100 mg·L^−1^ GA_3_ were applied to sweet cherry at the flowering stage, the soluble solids in the fruit increased by 7% and 12%, the hardness increased by 15% and 20%, and the weight increased by 7% and 14%, respectively, compared with the untreated group [12]. Einhorn et al.’s study showed that applying a low concentration (10–25 mg·L^−1^) of GA_3_ could continuously improve the fruit firmness of sweet cherry by 10–43% [13]. Kuhn et al.’s study found that GAs can delay the maturation of sweet cherry fruit by inhibiting the accumulation of abscisic acid (ABA) through *PavSnRK2s* and *Pavpp2c*, which are negative regulators of the ABA pathway [14]. In addition, the results of Ozturk et al.’s study showed that preharvest gibberellin (GA_3_) and calcium chloride (CaCl_2_) and postharvest modified atmosphere packaging (MAP) treatments could maintain a higher vitamin C content and stronger antioxidant activity of sweet cherry fruits during storage [15]. However, Correia et al. showed that the vitamin C content of sweet cherry fruit decreased after GA_3_ treatment, mainly caused by changes in local climate conditions and spraying time [16].

Recently, the effect of GAs on the fruit-setting rate and fruit quality of sweet cherries has been studied more intensively, but the effect of GAs on the formation mechanism of malformed sweet cherry fruit has not been reported in detail. In this study, the content of phytohormones in malformed and normal sweet cherry fruits after spraying GAs at the flowering stage was determined, and the differentially expressed genes were analyzed using transcriptome data and qRT-PCR. The results of this study will lay a foundation for research on the formation mechanism of malformed sweet cherry fruit.

## 2. Results

### 2.1. Metabolomics Analysis of Sweet Cherry

In order to detect the phytohormones of two different sweet cherry fruits, the multiple reaction monitoring (MRM) model was used to analyze the extracts of malformed fruit (T) and normal fruit (CK) (Figure 1 and Appendix A), in which a total of 20 phytohormones were detected (Table 1). The hormone content in malformed fruits was higher than that in normal fruit, except for abscisic acid (ABA) and indole-3-carboxaldehyde (ICAld). Among them, the most abundant are L-tryptophan (TRP), 1-aminocyclopropenecarboxylic acid (ACC), and gibberellin A_3_ (GA_3_), which belong to auxin, ethylene, and gibberellin, respectively. The TRP content of T increased by 7.7% compared with CK. The ACC content of T was 18% higher than that of CK. The most significant difference in plant hormone content was GA_3_. The GA_3_ content of T was seven times that of CK. The results showed that more plant hormones regulated malformed fruit growth and development, and GA_3_ played an important role in malformed fruits.

The principal component analysis (PCA) results of the metabolome profiles are shown in Figure 2. Six samples can be separated from the first two principal components, accounting for 59.82% and 21.69% of the total variation rate, respectively. In the PCA analysis chart, the two groups showed a separation trend, and each repeated cluster of the two groups showed good experimental data. The clustering results of group T were more concentrated than those of group CK, indicating that the repeatability of group T in the experiment was better than that of group CK. Significant differences existed between the two sweet cherry samples; the T group was distributed on the positive end of PC1, while the CK group was distributed on the negative end of PC1. In addition, in PC2, the T group was distributed on the positive end, and the CK group was distributed on the negative end.

### 2.2. Transcriptome Analysis of Sweet Cherry Fruit

Six RNA libraries (CK1, CK2, and CK3 for the normal fruit, and T1, T2, and T3 for malformed fruits) were prepared and analyzed. In total, 268.19 (M) raw reads were obtained from the six libraries. The raw reads in this study were deposited in the National Center for Biotechnology Information (NCBI) Sequence Read Archive (SRA) database (PRJNA1036423). For further analysis, after raw read quality filtering, 126.99 (M) and 125.80 (M) clean reads were obtained from the CK and T libraries, respectively, among which the average Total Clean Bases content was 6.32 Gb (Table 2). In addition, the Q30 percentages of all six libraries were above 85%, indicating that the sequencing and RNA qualities were high, and the data obtained were reliable enough for further profile studies on gene expression.

The expression of transcript samples was analyzed using PCA, and the results are shown in Figure 3A. This figure shows that each sample can be clearly distinguished on the score map, and the resets are closely focused, indicating that the transcripts of the two fruits are different. Similar to metabolomics, the CK group was located at the negative end of PC2, while the T group was located at the positive end of PC2. Interestingly, the score map shows that the difference between group CK and group T on PC1 is very weak, indicating that the two samples are similar. Moreover, the sample gene expression correlation between the replicates of each sample was the highest, indicating that the samples had good repeatability (Figure 3B). Differential gene screening was performed with a threshold of *p*-value < 0.05 and |log2FoldChange| > 2, and a total of 3573 different expression genes (DEGs) were identified, as shown in Figure 3C. Subsequently, we compared the differential expression of genes between normal fruit (CK) and malformed fruit (T) in sweet cherry. The volcanic map showed that 2137 genes were up-regulated and 1400 genes were down-regulated in 3573 DEGs (Figure 3D). In addition, hierarchical clustering heat maps of the two kinds of fruits were drawn (Figure 3E), and the changes in expression levels were represented by red, white, or blue gradients, respectively. Red represents a high expression level, and blue represents a low expression level. The results showed that the repeatability of each group was good, and the transcription differences between groups were large.

### 2.3. Combined Metabolome and Transcriptome Analysis

The Kyoto Encyclopedia of Genes and Genomes (KEGG) analysis provides information and further understanding of how sweet cherry regulates its biological functions and the biosynthesis of secondary metabolites at the transcriptome and molecular level. It has been shown that the DEGs are enriched in the 130 KEGG pathways, among which the most significantly enriched pathways were plant hormone signal transduction, tryptophan metabolism, indole alkaloid biosynthesis, diterpenoid biosynthesis, and carotenoid biosynthesis. These pathways are mainly composed of up-regulated DEGs (Figure 4). Among the 130 KEGG pathways, there is 1 pathway related to fruit shape—namely, the diterpenoid biosynthesis pathway (ko00904), with 14 unigenes. The pathways annotated by these DEGs were closely related to the abundance of metabolites of CK and T.

In the KEGG enrichment analysis, we found that the genes related to “diterpenoid biosynthesis” were significantly highly expressed, and the “GA biosynthesis process” pathway was a branch of the “diterpenoid biosynthesis” pathway (Figure 5A). In the “GA biosynthesis process” pathway, gibberellin-44 dioxygenase (EC: 1.14.11.12, *GA20ox*), gibberellin 2beta dioxygenase (EC: 1.14.11.13, *GA2ox*), and gibberellin 3beta dioxygenase (EC: 1.14.11.15, *GA3ox*) were up-regulated and down-regulated, respectively. Among them, the *GA20ox1* (LOC110745003), *GA20ox2* (LOC110762932), *GA20ox3* (LOC110762934), *GA20ox4* (LOC110765963), and *GA20ox5* (LOC110765965) genes were up-regulated in gibberellin-44 dioxygenase (EC: 1.14.11.12, *GA20ox*). In gibberellin 2beta dioxygenase (EC: 1.14.11.13, *GA2ox*), *GA2ox1* (LOC110754555) and *GA2ox2* (LOC110772934) were up-regulated, and *GA2ox3* (LOC110773997) was down-regulated. Finally, in gibberellin 3beta dioxygenase (EC: 1.14.11.15, *GA3ox*), *GA3ox* (LOC110750486) was up-regulated (Figure 5B, Appendix A). In this study, we observed that the regulation of GA biosynthesis-related genes may be related to the formation of malformed sweet cherry fruit. On the other hand, the metabolomic data showed that 20 different metabolites were enriched in 130 metabolic pathways. The differential genes and metabolites in the diterpenoid biosynthesis pathway were analyzed using metabolomics and transcriptomics. The results showed that the expression of gibberellin-related enzyme genes in malformed fruits was higher than in normal fruits.

### 2.4. Expression of Gibberellin-Related Synthase Gene

The transcriptional regulation revealed by RNA-seq was confirmed in the biological independent experiments of qRT-PCR, and the results of both were consistent. The positive correlation coefficient (R2) between the RNA-seq data and the qRT-PCR was 0.86 (Appendix A). In the process of GAs biosynthesis, we selected nine differentially expressed genes (*GA20ox1*, *GA20ox2*, *GA20ox3*, *GA20ox4*, *GA20ox5*, *GA2ox1*, *GA2ox2*, *GA2ox3*, and *GA3ox*) of three key metabolic enzymes (*GA20ox*, *GA2ox*, and *GA3ox*) in normal (CK) and malformed fruits (T) (Figure 6). Except for *GA2ox3*, the expression of all other genes showed much higher gene expression levels in T than in CK. Among them, the expression of all genes of *GA20ox* and *GA3ox* in T was significantly higher than that in CK. Among the genes involved in regulating the expression of gibberellin 2-dioxygenase (*GA2ox*), compared with CK, *GA2ox3* had no significant changes in T, while *GA2ox1* and *GA2ox2* were significantly expressed in T. The results showed that the genes of gibberellin-related enzymes were significantly expressed in malformed fruits, especially the up-regulated expression of gibberellin-related synthase (*GA20oxs* and *GA3oxs*), which led to the accumulation of GA_3_ in fruits. The increased expression level of gibberellin-degrading enzyme (*GA2oxs*) may be caused by the accumulation of GA content in fruit.

## 3. Discussion

Plant hormones are involved in a variety of metabolic activities. In order to explore the relationship between plant hormones and malformed fruits, we measured the hormone differences between normal fruits and malformed fruits and analyzed the transcriptome data. In our study (Table 1), sweet cherry fruit contained more plant hormones such as auxin (TRP, ICA, etc.), abscisic acid (ABA and ABA-GE), ethylene (ACC), and GAs (GA3, GA1, etc.). The auxin content in sweet cherry fruit was the highest, followed by ethylene, GAs, and abscisic acid. Auxin is a multifunctional plant hormone that is involved in almost all activities in the plant life cycle, especially in fruit development and fruit setting [17]. The exogenous application of auxin can delay the maturation of tomato fruit. Li et al. [18] showed that exogenous auxin can up-regulate the expression of genes related to stress resistance; down-regulate the genes related to carotenoid metabolism, cell degradation, and energy metabolism; maintain the high methylation level in the nucleus; and thus inhibit the ripening process of tomato. In addition, other studies have shown that the expression patterns of transcription factor genes related to maturation in tomatoes after auxin treatment are also affected by exogenous auxin [19]. Auxin/indole-3-acetic acid (Aux/IAA) and auxin response factor (ARF), which encode the transcriptional repressor and transcription factor, respectively, are key components of the auxin signaling pathway [20]. In *Brassica napus* L., the BnARF18 mutant restricted cell expansion in the cell wall, resulting in a decrease in seed size [21]. *Arf106* is expressed during cell division and cell expansion, which is an indirect factor controlling apple fruit size [22]. Ethylene regulation of fruit ripening involves multiple ethylene receptor proteins, transcription factors, and downstream target genes, forming a complex network [23]. The ethylene response factor is an important transcription factor in the ethylene signaling pathway, which regulates the transcription of fruit-ripening genes by binding to the AP2 site in the promoter [24]. In addition, auxin and ethylene interact to regulate fruit maturation through various physiological processes. In this study, sweet cherry contains a large amount of auxin and ethylene at the maturity stage, which verifies that fruit maturation requires the joint action of auxin and ethylene [25]. GAs are an important plant growth and development hormone, which is necessary for seed development and can induce seedless fruit and promote fruit development [26]. In GA signal transduction, the GA signal is sensed by GA intrinsic Dwarfs (GIDS) to form a ga-gid1-della complex, which changes the structure of the DELLA protein and causes downstream gene expression [27]. It is well known that auxin and GAs jointly regulate fruit development in the primary stage of fruit development. Auxin can induce the synthesis and activity of GAs, and GA accumulation can act on the DELLA protein and GA inhibitor, triggering the GA signal to start fruit growth [28,29]. In this study, the GA content of malformed fruit was much higher than that of normal fruit, and the main difference was the GA_3_ content, which may be the key factor in the malformed fruit of sweet cherry.

In this study, the candidate genes of malformed fruit were obtained using RNA-seq technology (Table 2, Figure 3). Among the KEGG enrichment pathways (Figure 4), we focused on five key pathways, namely, “plant hormone signal transduction”, “tryptophan metabolism”, “indole alkaloid biosynthesis”, “dieterpenoid biosynthesis”, and “carotenoid biosynthesis”. Plant hormones are involved in various physiological processes of plants, including growth, development and aging. Therefore, in order to ensure the normal growth process of plants, plant hormone signal transduction is very important [30]. The main plant hormones are auxin, GAs, cytokinin, abscisic acid, ethylene, and salicylic acid. In this study, 111 differential unigenes were enriched in phytohormone signaling pathways between malformed and normal fruits. It was further proved that many hormones regulate the maturation and development of malformed fruits, and the hormone content differs from that of normal fruits. Tryptophan metabolism and indole alkaloid biosynthesis are both related to auxin. Tryptophan is a common and important amino acid in root exudates. It is the main precursor of indole-3-acetic acid (IAA) biosynthesis in bacteria and higher plants [31], and IAA is one of the auxins we have studied most and used most widely. Studies have shown that plant-related bacteria use L-tryptophan as a substrate to synthesize IAA and promote the growth of various plants [32,33]. In our hormone content study, auxin content was the highest, indicating that auxin plays an important role in the development and maturation of sweet cherry fruit, which is consistent with previous results. In our study, the GA synthesis pathway was a part of the diterpene metabolism pathway, which is similar to the results of Sun and Kamiya [34]. Moreover, GAs can not only play a role in plant stress resistance and growth but also regulate fruit size. Therefore, we focused on the genes in the GA biosynthesis pathway. The precursor of GA synthesis in plants is geranylgeranyl diphosphate (GGPP), which forms two pathways of GA_12_ and GA_53_ under the action of a series of enzymes. Finally, GA20oxs, GA3oxs, and GA2oxs were oxidized in the cytoplasmic matrix to form different forms of GAs, of which only a few GAs (GA_1_, GA_3_, GA_4_, and GA_7_) had biological activity [35]. GA2oxs is a key enzyme in GA metabolism, which can inactivate the bioactive GA in plants, thus maintaining the balance of bioactive GA content in plants [36]. GA20oxs and GA3oxs are synthases in GA metabolism. The loss of function of GA20ox and GA3ox leads to the dwarfing phenotype [37,38]. In this study, the gene expression levels of GA20oxs, GA3oxs, and GA2oxs in malformed fruits were higher than those in normal fruits (Figure 5B and Figure 6), leading to the accumulation of bioactive GA_3_, and the GA_3_ content in malformed fruits was significantly higher than that in normal fruits. GA plays an important role in flower bud differentiation, and GA metabolism during flower bud differentiation depends on temperature [39]. In determining endogenous GA in *Phalaenopsis*, GA_3_ could be detected only in the early stage of flower bud differentiation [40], and the same result was also found in tomatoes. In this study, there is still a large amount of GA_3_ in the mature fruit of sweet cherry, and the content of malformed fruit is seven times higher than that of normal fruit. We speculate that the formation of malformed fruit may be related to the existence of GA_3_. However, how GA_3_ regulates the formation of malformed fruits needs further study.

## 4. Materials and Methods

### 4.1. Plant Materials

In this study, sweet cherry was selected as the experimental material, and the experimental site was located at the Tiangong Manor sweet cherry experimental base of Yinzhou district, Ningbo City, Zhejiang Province, China (E 121.58172°, N 29.80472°). Ten sweet cherry fruit trees with good growing conditions and the same developmental period were selected for testing. The entire tree was sprayed with GA3 (25 mg/L) meticulously in the early flowering period and 10 days after full bloom. A control group of normal fruit (CK) and a test group of malformed fruit (T) were set up in the experiment (Figure 1 and Appendix A). After 32 days of the second GA_3_ treatment, 3 normal fruits and 3 malformed fruits were picked from 10 experimental sweet cherry trees, respectively. After picking, the fruits were brought back to the laboratory immediately, then frozen in liquid nitrogen, and stored in a −80 °C refrigerator for subsequent experiments. Three biological replicates were prepared for each sample.

### 4.2. Metabolite Extraction and Profiling

The sample was pulverized, and a 50 μL sample was mixed with 1 mL methanol/water/formic acid (15:4:1, *v*/*v*/*v*). A total of 10 μL of internal standard mixed solution (100 ng/mL) was added into the extract as an internal standard (IS) for the quantification. The mixture was vortexed for 10 min, and then centrifuged for 5 min (12,000 r/min, 4 °C). The supernatant was then transferred to clean plastic microtubes, followed by evaporation to dryness, dissolved in 100 μL of 80% methanol (*v*/*v*), and filtered through a 0.22 μm membrane filter for further UPLC-MS/MS analysis [41,42].

The sample extracts were analyzed using a UPLC-ESI-MS/MS system (UPLC, ExionLC™ AD, Sciex, Framingham, MA, USA, https://sciex.com.cn/ (accessed on 13 April 2022); MS, Applied Biosystems 6500 Triple Quadrupole, Sciex, Framingham, MA, USA, https://sciex.com.cn/ (accessed on 13 April 2022)). The analytical conditions were as follows: chromatographic column: Waters ACQUITY UPLC HSS T3 C18 (100 mm × 2.1 mm i.d., 1.8 µm); solvent system: water with 0.04% acetic acid (A), acetonitrile with 0.04% acetic acid (B); gradient program: started at 5% B (0–1 min), increased to 95% B (1–8 min), 95% B (8–9 min), and finally ramped back to 5% B (9.1–12 min); flow rate: 0.35 mL/min; temperature: 40 °C; injection volume: 2 μL [43,44,45]. Linear ion trap (LIT) and triple quadrupole (QQQ) scans were acquired on a triple quadrupole-linear ion trap mass spectrometer (QTRAP), QTRAP^®^ 6500+ LC-MS/MS System, equipped with an ESI Turbo IonSpray interface, operating in both positive and negative ion mode and controlled using Analyst 1.6.3 software (Sciex). The ESI source operation parameters were as follows: ion source, ESI+/−; source temperature 550 °C; ion spray voltage (IS) 5500 V (Positive), −4500 V (Negative); and curtain gas (CUR) was set at 35 psi, respectively. Phytohormones were analyzed using scheduled multiple reaction monitoring (MRM). Data acquisitions were performed using Analyst 1.6.3 software (Sciex). Multiquant 3.0.3 software (Sciex) was used to quantify all metabolites. Mass spectrometer parameters, including the declustering potentials (DP) and collision energies (CE) for individual MRM transitions, were created with further DP and CE optimization. A specific set of MRM transitions was monitored for each period according to the metabolites eluted within this period [46,47,48].

### 4.3. RNA Extraction and RNA-Sequencing

After 32 days of the second GA_3_ treatment, 3 normal fruits and 3 malformed fruits were picked from 10 experimental sweet cherry trees. The RNA was extracted from the fruit after mixed samples were ground into powder. Ethanol precipitation protocol and CTAB-PBIOZOL reagent were used to purify total RNA from the plant tissue according to the manual instructions. The ground fruit powder sample was put into 1.5 mL of CTAB-pBIOZOL reagent preheated at 65 °C. The samples were incubated by Thermo mixer for 15 min at 65 °C to permit the complete dissociation of nucleoprotein complexes. After centrifuging at 12,000× *g* for 5 min at 4 °C, 400 µL of chloroform per 1.5 mL of CTAB-pBIOZOL reagent was added to the supernatant, and it was centrifuged at 12,000× *g* for 10 min at 4 °C. The supernatant was transferred to a new 2.0 mL tube to which 700 µL of acidic phenol and 200 µL of chloroform were added, followed by centrifuging 12,000× *g* for 10 min at 4 °C. A total of 400 µL of chloroform was added to the supernatant and it was centrifuged at 12,000× *g* for 10 min at 4 °C. An equal volume of isopropyl alcohol was added to the supernatant, and it was kept steady at −20 °C for 2 h for precipitation. After that, the mix was centrifuged at 12,000× *g* for 20 min at 4 °C, and then the supernatant was removed. After being washed with 1 mL of 75% ethanol, the RNA pellet was air-dried in the biosafety cabinet and was dissolved by adding 50 µL of DEPC-treated water. Subsequently, the total RNA was qualified and quantified using a Nano Drop and Agilent 2100 bioanalyzer (Thermo Fisher Scientific, Waltham, MA, USA) [49].

The library was established and sequenced via BGI (BGI Genomics Co., Ltd., Shenzhen, China) using the BGIseq500 platform (BGI-Shenzhen, Shenzhen, China) [50]. The sequencing data were filtered with SOAPnuke [51] by (1) removing reads containing sequencing adapter, (2) removing reads whose low-quality base ratio (base quality less than or equal to 15) was more than 20%, and (3) removing reads whose unknown base (“N” base) ratio was more than 5%. Afterward, clean reads were obtained and stored in FASTQ format. The subsequent analysis and data mining were performed on the Dr. Tom Multi-omics Data mining system (https://biosys.bgi.com, (accessed on 4 December 2021)).

### 4.4. Combined Metabolome and Transcriptome Analysis

Significantly regulated metabolites between groups were determined by absolute Log2FC (fold change). The identified metabolites were annotated using the KEGG compound database (http://www.kegg.jp/kegg/compound/, (accessed on 13 April 2022)), and the annotated metabolites were then mapped to the KEGG Pathway database (http://www.kegg.jp/kegg/pathway.html, (accessed on 13 April 2022)). Pathways with significantly regulated metabolites mapped were then fed into MSEA (metabolite sets enrichment analysis), and their significance was determined via the hypergeometric test’s *p*-values.

### 4.5. Different Expression Gene (DEG), KEGG, and Gene Ontology (GO) Enrichment Analysis

HISAT2 (v2.1.0) [52] and Bowtie2 (v2.3.4.3) [53] were applied to align the clean reads to the gene set, in which known and novel and coding and noncoding transcripts were included. The expression level of genes was calculated using RSEM (v1.3.1) [54]. The heatmap was drawn using pheatmap (v1.0.8) according to the gene expression difference in different samples. Essentially, a differential expression analysis was performed using the DESeq2 (v1.4.5) [55] (or DEGseq [56] or PoissonDis [57]) with Q value ≤ 0.05 (or false discovery rate (FDR) ≤ 0.001).

To gain insight into the change of phenotype, GO (http://www.geneontology.org/, (accessed on 28 January 2022)) and KEGG (https://www.kegg.jp/, (accessed on 28 January 2022)) enrichment analysis of annotated differentially expressed genes was performed using Phyper (https://en.wikipedia.org/wiki/Hypergeometric_distribution, (accessed on 28 January 2022)) based on the hypergeometric test. The significant levels of terms and pathways were corrected by Q value with a rigorous threshold (Q value ≤ 0.05).

### 4.6. Quantitative Real-Time Reverse Transcription PCR (qRT-PCR) Analysis

The total RNA of experimental materials was extracted using the Polysaccharide/Polyphenol Plant RNA Rapid Extraction Kit (gDNA Clearance Column) (HLingene, Shanghai, China), and fluorescent cDNA was synthesized according to the instructions for the NovoScript^®^Plus All-in-one 1st Strand cDNA Synthesis SuperMix (gDNA purge) (Novoprotrin, Shanghai, China). The qRT-PCR analysis was performed according to the instructions for the Novostart^®^ SYBR qPCR SuperMix Plus (Novoprotrin, Shanghai, China) kit. The reaction system included 35 μL of 2 × Novostart^®^ SYBR qPCR Supermix plus, 1.4 μL each of the forwards and reverse primers, 1.4 μL of fluorescent cDNA, 30.8 μL of ddH_2_O, and a total volume of 70 μL. The reaction procedure was as follows: predenaturation at 95 °C for 5 min, followed by 30 cycles of 95 °C for 1 min, 95 °C for 20 s, and 60 °C for 1 min. The relative expression level of the genes was determined using the 2^−ΔΔCT^ method, using β-actin as the respective reference gene. The primers used for qRT-PCR are listed in the Appendix A.

### 4.7. Statistical Analysis

Each result needs to be repeated three times to reduce experimental error. Metabolome data statistics and transcriptome sequencing were prepared on BGI (Wuhan, China), respectively. For the statistical evaluation of the metabolome and transcriptome, the mean value and standard deviation of the three replicates were calculated, and the significant differences between groups were evaluated via a one-way analysis of variance (ANOVA) at *p* < 0.05 using SPSS 25.0 software version. The graphs were plotted using Graphpad prism 8.0 and Origin 2023.

## 5. Conclusions

In this study, UPLC-MS, RNA-seq, and qRT-PCR were used to preliminarily explore the causes of the formation of malformed sweet cherry fruit. There was a significant difference in GA_3_ content between malformed and normal fruit. The RNA-seq sequencing results of malformed fruits showed that the differentially expressed genes were significantly enriched in the KEGG pathway, including “plant hormone signal transduction”, “diterpenoid biosynthesis”, and “carotenoid biosynthesis”. The further analysis of the expression levels of GA-related synthesis and metabolism genes showed that GA hydrolase gene *GA2oxs* and GA synthase genes *GA20oxs* and *GA3oxs* were significantly up-regulated. In conclusion, we speculate that the expression of *GA20oxs*, *GA3oxs*, and *GA2oxs* is significantly up-regulated after exogenous GA_3_ spraying, leading to the existence of a large amount of GA_3_ in the fruit from flowering to fruit maturity, which leads to malformed sweet cherry fruit. However, it is unclear whether GA_3_ interacts with other plant hormones to regulate the formation of malformed fruits and which transcription factors are involved, and these questions require further research.

## Figures and Tables

**Figure 1 ijms-25-00153-f001:**
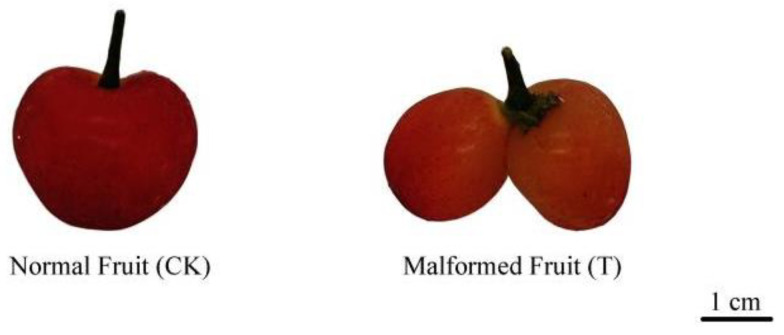
Normal fruit and malformed fruit of southern sweet cherry.

**Figure 2 ijms-25-00153-f002:**
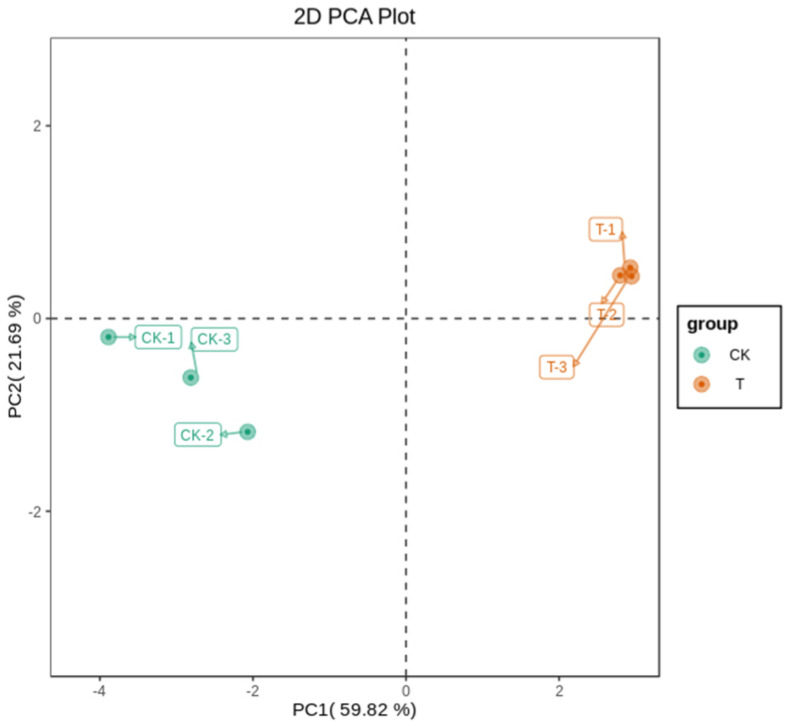
Differential hormone metabolite analysis based on principal component analysis (PCA). The horizontal coordinates represent principal component 1 and the vertical coordinates represent principal component 2. Different colored dots represent different groups of samples.

**Figure 3 ijms-25-00153-f003:**
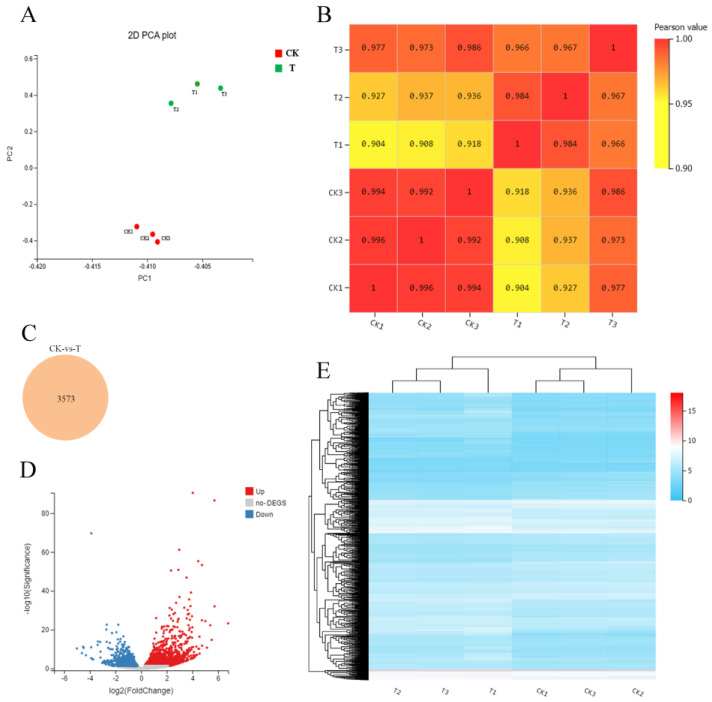
Transcriptome analysis of sweet cherry fruits. (**A**) PCA analysis of gene expression in sweet cherry fruit. (**B**) Spearman correlation coefficient of gene expression in sweet cherry fruit. (**C**) Venn diagram of DEGs in sweet cherry fruit. (**D**) Volcanic map of DEGs in sweet cherry fruit. (**E**) Hierarchical clustering of DEGs in sweet cherry fruit.

**Figure 4 ijms-25-00153-f004:**
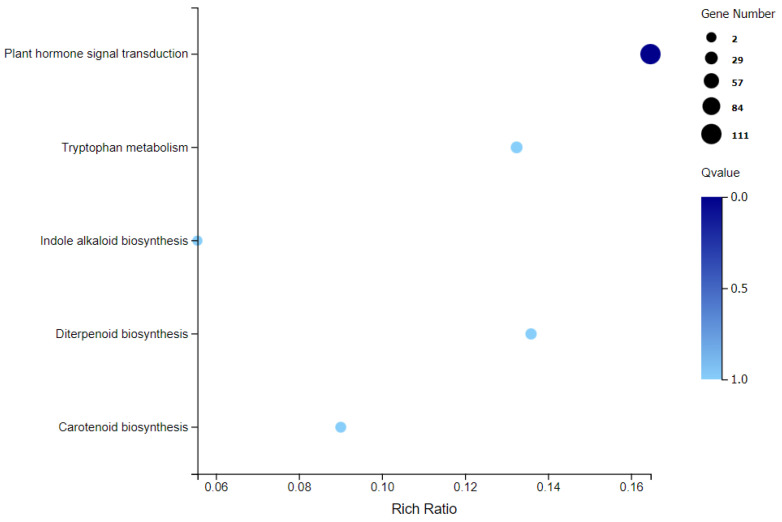
Scatterplot of the significantly enriched KEGG pathways in CK and T.

**Figure 5 ijms-25-00153-f005:**
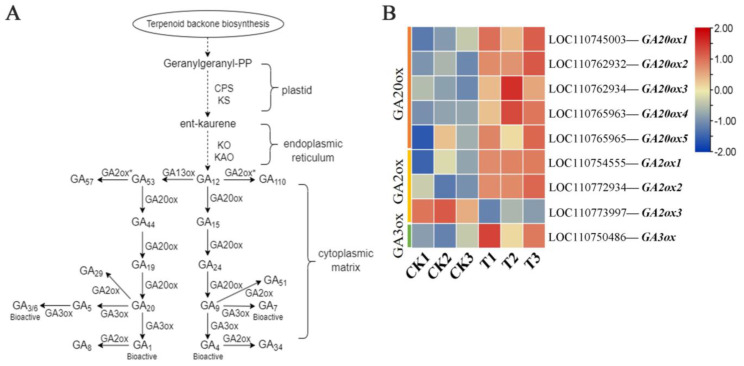
(**A**) The biosynthetic pathway of gibberellin (GA) in sweet cherry. The GA pathway is based on the study of Chen et al. [4]. (**B**) Visual heatmap of gene expression of gibberellin-related enzymes in the fruit of sweet cherry. GA2ox*: gibberellin-related synthetase; GA2ox: gibberellin-degrading enzyme.

**Figure 6 ijms-25-00153-f006:**
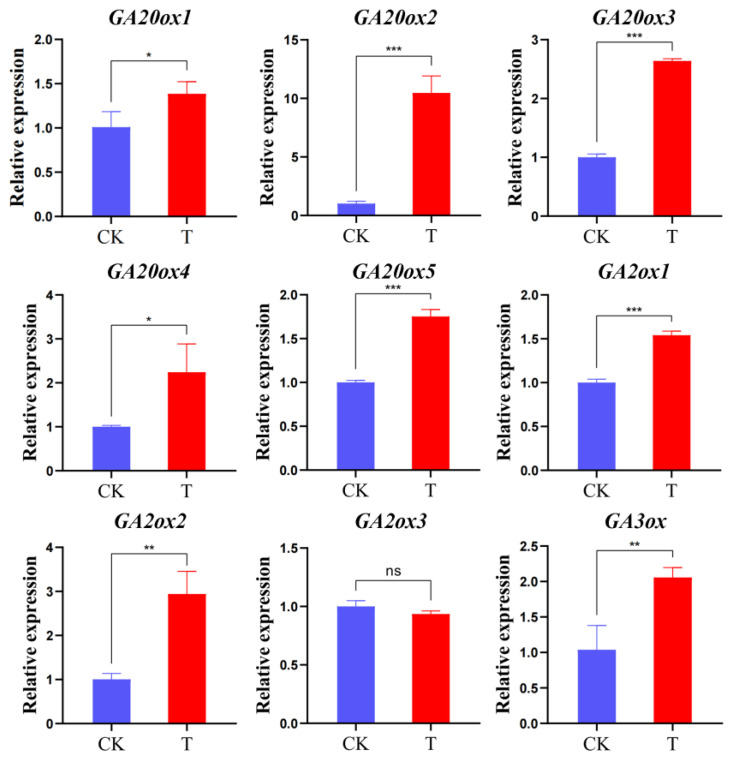
Expression patterns of gibberellin-related enzyme genes in fruit ripening stage using qRT-PCR. Data were normalized against an internal control β-actin, and the expression level at CK was set as 1. Error bars indicate standard deviation, and asterisks indicate significant differences between the CK and T. * *p* < 0.05, ** *p* < 0.01, *** *p* < 0.001, and ns: no significant difference.

**Table 1 ijms-25-00153-t001:** Type and content of plant hormones in normal fruit (CK) and malformed fruit (T).

Class	Index	Compounds	CK (ng/g)	T (ng/g)
ABA	ABA	Abscisic acid	12.97 ± 2.15	12.23 ± 0.19
ABA	ABA-GE	ABA-glucosyl ester	42.20 ± 6.31	49.68 ± 1.23
Auxin	IAA-Glu	Indole-3-acetyl glutamic acid	UD	0.30 ± 0.04
Auxin	IAA-Asp	Indole-3-acetyl-L-aspartic acid	UD	2.88 ± 0.52
Auxin	TRA	Tryptamine	0.09 ± 0.04	1.81 ± 1.65
Auxin	IPA	3-Indolepropionic acid	0.09 ± 0.01	0.15 ± 0.02
Auxin	ICA	Indole-3-carboxylic acid	8.13 ± 2.68	10.01 ± 1.65
Auxin	TRP	L-tryptophan	3297.76 ± 81.85	3552.37 ± 97.42
Auxin	MEIAA	Methyl indole-3-acetate	0.27 ± 0.05	4.37 ± 0.34
Auxin	ICAld	Indole-3-carboxaldehyde	12.94 ± 2.25	12.73 ± 0.34
Auxin	IAN	3-Indoleacetonitrile	0.19 ± 0.02	0.27 ± 0.02
Auxin	IAA-Val-Me	Indole-3-acetyl-L-valine methyl ester	0.01 ± 0.00	0.01 ± 0.00
Auxin	IAA	Indole-3-acetic acid	1.84 ± 0.24	9.25 ± 7.15
ETH	ACC	1-Aminocyclopropanecarboxylic acid	177.57 ± 11.87	210.49 ± 3.47
GA	GA_20_	Gibberellin A_20_	0.57 ± 0.06	1.75 ± 0.54
GA	GA_4_	Gibberellin A_4_	0.18 ± 0.02	0.98 ± 0.03
GA	GA_7_	Gibberellin A_7_	0.04 ± 0.01	0.24 ± 0.01
GA	GA_3_	Gibberellin A_3_	55.49 ± 2.57	378.21 ± 32.27
GA	GA_1_	Gibberellin A_1_	34.91 ± 4.54	35.02 ± 4.25
GA	GA_19_	Gibberellin A_19_	0.69 ± 0.01	1.71 ± 0.12

Note: UD means that the substance was under-detected.

**Table 2 ijms-25-00153-t002:** Summary of the sequencing quality of six RNA libraries of sweet cherry.

Sample	Total Raw Reads (M)	Total Clean Reads (M)	Total Clean Bases (Gb)	Clean Reads Q20 (%)	Clean Reads Q30 (%)	Clean Reads Ratio (%)
CK1	43.82	42.45	6.37	97.84	93.62	96.87
CK2	43.82	42.43	6.36	97.78	93.39	96.82
CK3	45.44	42.11	6.32	96.45	89.48	92.67
T1	45.44	42.49	6.37	96.62	89.86	93.50
T2	45.44	42.45	6.37	96.34	89.04	93.43
T3	44.23	40.86	6.13	96.45	89.51	92.38

## Data Availability

All data generated or analyzed during this study are included in the Appendix A. The raw RNA-seq data are freely available at https://www.ncbi.nlm.nih.gov/sra/PRJNA1036423 (accessed on 16 November 2023). There are no ethical issues involved in this work. There was no experimentation with living animals in this work.

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
