# Peer review of "Preliminary Study on the Formation Mechanism of Malformed Sweet Cherry (Prunus avium L.) Fruits in Southern China Using Transcriptome and Metabolome Data"

_ijms, 2023, doi:10.3390/ijms25010153_

Round 1
Reviewer 1 Report
Comments and Suggestions for Authors
The manuscript entitled ’’Preliminary Study on the Formation Mechanism of Sweet Cherry Malformed Fruits in Southern China Using Transcriptome and Metabolome Data’’ presented by Xu et al., is of standard quality of a scientific paper. The study dealt with an important phenomenon from both production and plant physiology perspective. The article is written clearly, well structured, and an adequate number of analysis have been performed to process the data. In my view the work can be accepted for publication in IJMS. Some minor issues are listed below to help the authors for further improvements.
Abstract and elsewhere
Line 18: correct ‘’GA3’’ to ‘GA3’
Line 21: write the gene names in Italic. ‘’GA2Ox’’ to ‘GA2Ox’
Line 55: add the name of the plant when bringing such statements: ‘’of the fruit [10]’’ to ’’of the apple fruit [10]’
Line 61: complete the sentience: ‘’…through PavSnRK2s and Pavpp2c’’ to ‘through PavSnRK2s and Pavpp2c …???…’
Line 66: Remove the dot (.) before [16]
Line 164: Give the names of the selected primer sequences of targeted genes and the internal standard gene in the Table S1
Line 172: Give a reference for the Figure 5A. Like adopted or reproduced from xxx et al
Line 184: correct the sentence: ‘’In the gene of GA2ox, 110773997 gene was slightly 184
down-regulated…’’
Line 186: what does it mean by ‘’genes were larger’’?
Line 192: give the name of the genes’ abbreviation instead of the numbers on charts
Line 200: which fruit? ‘’the fruit contains more plant hormones’’ the same for line: 203
Line 203: How is it relevant to your study? : ‘’Anton et al. [17] found that the content of auxin was the highest in apple fruit, which was consistent with the results of this study. However, cytokinin, salicylic acid and jasmonic acid were also detected in apple, and the content of cytokinin was second only to auxin content, while ethylene content was not detected.’’
Line 206: Not relevant: ‘’In addition, the hormone content in fruit pulp is also different from that in fruit juice.’’ delete this kind of text and discuss the results better with studies that are more relevant.
Line 318. Clearly describe the plant sampling for RNA and transcriptome analysis. Days after treatment, number of biological samples form one or more individual plants, which plant tissue? And so on…Also please upload the raw RNA seq data in some available databases and give the Bio project accession umbers
Line 323-333: Rephrase the sentences like: ‘’…the supernatant was added 400ul of chloroform’’ or ‘’… The aqueous phase was added equal volume…’’
Line 364: Give the names of the selected targeted genes and the internal standard gene for qRT-PCR in the Table S2
Reviewer 2 Report
Comments and Suggestions for Authors
Dear Editor,
I tried to read the manuscript but english language is very poor and it is very difficult to follow while reading. Moreover, the use of acronyms is not appropriate. Two examples for everyone: what do KEGG and DEG stand for? These acronyms are repeated many times along the manuscript but no explanation of their meaning is reported! By this way, it is very difficult to understand the relevance and significance of the reported results and conclusions.
By the way, in the introduction section most of the relevant literature on the subject is not reported. For example, it is widely known that gibberellin treatment positively influences cherry fruit quality (fruit size, fruit firmness, fruit acidity, vitamin-C, TSS, total sugars) and yield and none of these evidences is reported.
Therefore, I recommend to reject the manuscript in its current form, suggesting to the authors to critically revise the text and improve the clarity of presentation of their results to support their conclusions.
Comments on the Quality of English LanguageEnglish language is very poor. Many grammar errors are present along the manuscript and also the syntax of sentences is confusing. Extensive english language proofreading is recommended
Author Response
Dear reviewer,
We have made comprehensive revisions to our manuscript in response to the reviewer's comments. Please see the attachment.

Round 2
Reviewer 2 Report
Comments and Suggestions for Authors
Dear Editor, the authors have revised the manuscript and now the clarity of presentation is improved. The main aim of the manuscript and the achieved results are now well argumented. English language still requires moderate proofreading. I recommend the revision by a english speaking reviewer
Comments on the Quality of English LanguageEnglish language still requires moderate proofreading